# Utility of a Questionnaire Short Form for Adolescents with Listening Difficulties in Japan

**DOI:** 10.3390/children11101212

**Published:** 2024-10-02

**Authors:** Chie Obuchi, Yuka Sasame, Yayoi Yamamoto, Michiko Ashitani

**Affiliations:** 1Institute of Human Sciences, University of Tsukuba, Ibaraki 305-8572, Japan; 2Department of Speech and Hearing Sciences, International University of Health and Welfare, Chiba 286-8686, Japan; yyasuda@iuhw.ac.jp (Y.S.); yyamamoto@iuhw.ac.jp (Y.Y.); 3Faculty of Education, Shiga University, Shiga 520-0862, Japan; amichiko@edu.shiga-u.ac.jp

**Keywords:** listening difficulties, auditory processing disorder, listening problem, questionnaire, screening, adolescent

## Abstract

Background: A short and easy questionnaire is needed to identify symptoms of listening difficulties (LiD) at an early stage. This study aimed to evaluate the utility of such a questionnaire for adolescents with and without LiD. Methods: We included adolescents with and without LiD and adults without LiD in this study. We utilized a questionnaire designed for individuals with LiD, which combined the Speech, Spatial, and Quality of Hearing scales (SSQ)-12 and four additional psychological items. From this, we selected three items that exhibited the largest differences between adolescents with and without LiD. We subsequently examined the relationship between the total scores of all items and the three selected items to develop the short questionnaire. Results: The responses of adolescents to the questionnaire were consistent with those of adults. The total scores from the three selected items exhibited a strong correlation with the overall questionnaire score for adolescents both with and without LiD, indicating their potential for identifying LiD symptoms. Conclusions: The wide implementation of the short questionnaire developed in this study could lead to the early detection of potential LiD symptoms and timely intervention.

## 1. Introduction

Hearing and listening are important for children to develop their language skills. If children have hearing problems in everyday life, they may show developmental delays in language acquisition, communication, social skills, and quality of life [1,2,3,4]. Newborn babies are screened using newborn hearing screening tests immediately after birth. A recent study [5] reported that the prevalence of congenital hearing loss was 1.62 per 1000 newborns (bilateral, 0.84; unilateral, 0.77). The early hearing detection and intervention program is crucial for infants with hearing loss and their parents [6] and is conducted in many countries. The program follows the 1–3–6 goals: All infants should be screened for hearing loss before 1 month of age, diagnostic testing must be conducted before 3 months of age for infants who do not pass screening, and early intervention services before 6 months of age to detect permanent hearing loss [7].

Hearing loss is detected soon after birth using newborn hearing screening tests or physical examination and is also examined in relation to genetic auditory syndromes [8]. However, it is often difficult to detect symptoms of listening difficulties (LiD) at an early stage. Regarding the prevalence of LiD, Hind et al. reported that children and adults with LiD, despite normal audiograms, represent approximately 0.5–1.0% of the general population [9]. Another study indicated a prevalence of 1.94 per 1000 school-aged children [10]. Furthermore, individuals with LiD have normal audiograms but experience difficulties listening to speech in noisy situations, such as when several people are talking. This symptom was previously identified as auditory processing disorder (APD) [11,12]; however, the characterization of symptoms in LiD has recently been refined. Research suggests that individuals exhibiting LiD symptoms with normal audiograms should be diagnosed with LiD owing to their inadequate cognitive abilities required for effective listening [13,14,15,16]. De Wit et al. examined 48 studies through systematic review, and also reported that the listening difficulties of children or adolescents with APD may be a consequence of cognitive, language, and attention issues rather than bottom-up auditory processing [17]. A clear delineation of listening symptoms is essential [18]. However, the debate over terminology persists, with some studies employing the term APD and others favoring LiD. Wilson noted that both historical and current definitions of APD are nominal, characterized by being both stipulative (offering explicit and arbitrary definitions of word relationships) and operational (establishing rules that dictate how terms should be applied in specific cases), thus advocating for a conceptual framework for APD terminology [19]. Following Wilson’s rationale, we used “LiD” in this study, focusing on identifying symptoms indicative of listening difficulties rather than their etiology.

Basic audiological tests do not detect LiD symptoms because individuals may not always exhibit symptoms, and their difficulties depend on the listening environment. More complex auditory tasks, such as dichotic listening tests and speech-in-noise tests, are required to identify their symptoms [20]. However, these tests are typically performed when the symptoms become severe and necessitate further assessment, making early detection challenging. A simplified questionnaire could effectively detect LiD symptoms before their onset. Several questionnaires have been utilized for the differential diagnosis of LiD [21,22,23,24]. The Fisher’s Auditory Problems Checklist, a long-used questionnaire [21], includes 25 questions spanning categories such as acuity, attention, attention span, figure-ground perception, discrimination, short-term memory, long-term memory, comprehension, speech and language issues, auditory-visual integration, motivation, and performance. It serves as a useful screening tool for listening problems [25]. More recently, the auditory processing domains questionnaire (APDQ) has been applied in clinical settings [22], where parents assess their child’s frequency of competent performance across 52 questions divided into three scales: auditory processing, attention, and language. This study compared the parent’s rating results of the APDQ in a younger group (7–10 years) and an older group (11–17 years) with APD, attention deficit hyperactivity disorder, and learning disabilities. The result showed significant differences between normal and clinical groups for all scales.

The Speech, Spatial, and Quality of Hearing scales (SSQ) [24], primarily used for adults with hearing loss [26] and listening difficulties [25], have recently been adapted for children [27,28]. Falzone et al. [27] observed that questionnaires completed by children displayed an age effect, with children younger than 10 years old scoring lower than their older counterparts. Consequently, they recommended paying particular attention to children under 10 years old when administering the SSQ to this age group. Another study [29] examined the results of the LiD questionnaire in school-aged children and adolecents; the study indicated that the LiD score on a subjective questionnaire increased with school years and significantly differed from scores reported by guardians, with younger children being less likely to be aware of LiD. For effective early intervention of LiD, symptoms must be identified in childhood rather than adulthood using an appropriate questionnaire. Although these questionnaires effectively identify listening symptoms, they each comprise numerous questions.

The questionnaires currently employed [21,25,28,29] are used to identify school-aged children and adolescents suspected of having LiD and gauge the severity of such symptoms. However, they are not commonly utilized for symptom screening. If school audiologists are assigned to be present in schools, they can identify students with LiD symptoms. However, if audiologists are not available, early detection becomes difficult. In these scenarios, regular physical examinations are crucial for the early detection and timely intervention of LiD. These examinations aim to assess the general health of the population, prevent diseases, and promote well-being. During these assessments, physical measurements, medical history, and questionnaires are employed to identify signs of the disease. The questionnaire items are strategically selected to reduce the number of questions, facilitating rapid and effective symptom screening of LiD with fewer questions. However, no short questionnaires currently exist for screening LiD during physical examinations. Furthermore, there are few studies on the detection and examination of LiD symptoms in adolescents. 

In the present study, we aimed to examine the utility of a questionnaire designed for LiD for adolescents with and without LiD. Moreover, we investigated whether a short questionnaire for screening LiD symptoms could predict the overall score of the comprehensive questionnaire used in this study.

## 2. Materials and Methods

### 2.1. Participants

Adolescents with LiDs were recruited from a peer support organization for individuals with LiDs in Japan. Similarly, adolescents without LiDs were enlisted from general junior and senior high schools across Japan. Data collection spanned from June 2022 to March 2024. In this study, we included 180 adolescents aged 12 to 18 years (mean age 14.4 ± 1.1 years) who had normal hearing and no complaints of LiDs in daily life, alongside 55 adolescents aged 12 to 18 years (mean age 16.4 ± 1.4 years) who also had normal hearing but reported issues related to LiDs in everyday situations. These participants met the diagnostic criteria for LiD in Japan and were either referred or voluntarily joined the peer support organization. Adolescents without LiDs were selected from several cooperating schools. Based on previous studies [28,29], participants were limited to junior and senior high school students. These students reported no hearing problems during their annual physical examinations and were not aware of any LiD. After obtaining informed consent, they completed the questionnaire. We included responses from several junior and senior high school students who answered all questions. None of the participants had hearing loss, otorhinolaryngological abnormalities, intellectual disabilities, or a history of neurological disorders, head trauma, or surgery.

Adults without LiD from a previous study [30] were included in this study to examine the utility of the questionnaire for adolescents. The group comprised 58 adults aged 19 to 25 years (mean age 19.7 ± 0.83 years) who had normal hearing and no complaints of LiDs and whose total scores exceeded the questionnaire’s cutoff score.

This study received approval from the Ethics Committee of the International University of Health and Welfare in Chiba Prefecture, Japan (reference number: #2021-Im-001; ethical approval date: 27 April 2021) and was conducted in compliance with the Helsinki Declaration guidelines. All participants and their guardians provided informed consent and received a briefing prior to the study’s commencement.

### 2.2. Methods

We used a questionnaire for individuals with LiDs [30] that comprised a combination of the SSQ-12 [31] and four psychological items from the Questionnaire on Hearing 2002 [32] to assess the psychological aspects of LiD. The SSQ-12 is a refined 12-item questionnaire of the SSQ [24,33] for assessing a range of hearing disabilities across several domains in individuals with hearing impairments. The SSQ includes three domains: speech, spatial, and hearing qualities. The first domain of the SSQ, speech hearing, covers aspects such as competing sounds and the number of individuals involved in the conversations. For example: “You are talking with one other person and there is a TV on in the same room. Without turning the TV down, can you follow what the person you’re talking to says?” The second domain covers three components of spatial hearing: direction, distance, and movement. Finally, the third domain consists of items associated with issues such as segregation of sounds, recognition, clarity/naturalness, and listening effort. The Cronbach’s alpha coefficients of each domain were reported as follows: the speech domain was 0.97, the spatial domain was 0.94, the hearing qualities domain was 0.88, and the total score was 0.94 [34]. As mentioned above, the SSQ-12 comprises 12 of the original 49 SSQ items, tailored for clinical research and rehabilitation settings [31]. These 12 items were selected based on their importance in a clinical context, as independently nominated by three centers involved in the study project, drawing on their collective experiences with the SSQ.

Moreover, we incorporated four items from the 2002 Questionnaire on Hearing [32] into the LiD questionnaire [30], which included two domains: behavioral and emotional reactions, assessed by two and three items, respectively. We excluded one item from the emotional reactions domain due to its redundancy with another item. Additionally, four items from the psychological and social domains were employed. For example: “Do you think that you do not want to talk with others because of your listening difficulties?”. The 16 items are scored on a scale ranging from 0 to 10, with responses varying from “not at all” to “perfect”. The total scores ranged from 0 to 160. Obuchi and Kaga [30] employed this modified questionnaire in a survey among adults with and without LiD, analyzing the data using receiver operating characteristic analyses to establish a suitable cutoff score for LiD. The cutoff value for LiD in the questionnaire was ≤109. The findings indicated that this questionnaire is both highly sensitive and specific, and its efficacy was notably enhanced when SSQ-12 items were combined with those from the psychological domain [30].

We administered this questionnaire to adolescents with and without LiD. The questionnaire was self-administered and required individual responses. The participants were asked to answer each item based on listening status. We calculated the mean score for each item. Subsequently, we compared the results of the questionnaire for LiD between adolescents without LiD and adults without LiD from a previous study [30] to examine the utility of the questionnaire to adolescents. In addition, we selected three question items for screening LiD based on the largest differences observed between adolescents with and without LiD. After selecting these items, we examined the relationship between the total scores of all items and the three selected items to develop the short questionnaire.

### 2.3. Statistics

The internal consistency of the questionnaire in this study was measured by Cronbach’s alpha coefficient. Mean scores and standard deviations were calculated for each item in each group. The Shapiro–Wilk test was used to check for normality and parametric tests were employed based on that result. We compared the results of the participant groups and items using two-way analysis of variance (ANOVA) with the Scheffé test for post-hoc comparisons when ANOVA indicated significance. In addition, *t*-tests were conducted between participant groups.

Furthermore, three items with large differences between the groups were selected. We ensured that the domains of the items did not overlap. The relationship between the total scores of the full and three items was examined using Pearson’s correlation. Statistical significance was set at *p* < 0.01. All statistical analyses were conducted using the bell curve in Excel v.2.00 (Social Survey Research Information Co., Ltd., Tokyo, Japan).

## 3. Results

### 3.1. Comparison of the Results for Adolescents and Adults without LiD

Regarding the utility of the questionnaire for adolescents, the comparison between adolescents and adults without LiD from a prior study is presented in Table 1. The mean score for adolescents without LiD was 137.96 (SD 14.14) compared to 134.48 (SD 13.66) for adults without LiD. ANOVA revealed statistically significant differences between the participant groups (F = 12.68, *p* < 0.001), responses to the question items (F = 28.19, *p* < 0.001), and a significant interaction effect (F = 3.20, *p* < 0.001). There was a significant difference between participant groups only in the psychological domain using a *t*-test (total items: t = 1.64, *p* = 0.10, SSQ-12: t = 0.51, *p* = 0.61, psychological domain: t = 4.18, *p* < 0.001).

According to the Scheffé test for post-hoc comparisons, the only item that differed among the subject groups was item 15 (F = 51.23, *p* < 0.001). Furthermore, the correlation between age and the total questionnaire score for adolescents without LiD was not significant (r = −0.059, *p* = 0.42). These results indicate that the responses of adolescents to the questionnaire were similar to those of adults.

### 3.2. Comparison of the Results for Adolescents with and without LiD

The Cronbach’s alpha coefficient of the total items was 0.76, that of the SSQ-12 was 0.76, and that of the psychological domain was 0.71 in adolescents with LiD. The Cronbach’s alpha coefficient of the total items was 0.82, that of the SSQ-12 was 0.82, and that of the psychological domain was 0.70 in adolescents with LiD.

Table 2 presents the mean scores and standard deviations for each item in adolescents with and without LiD. The mean score of adults with LiD was 64.47 (SD 19.44). ANOVA revealed statistically significant differences in the participant groups (F = 3731.21, *p* < 0.001), responses to the question items (F = 35.31, *p* < 0.001), and a significant interaction effect (F = 27.71, *p* < 0.001). There was also significant difference between participant groups using the *t*-test (total items: t = 30.48, *p* < 0.001, SSQ-12: t = 23.85, *p* < 0.001, psychological domain: t = 28.98, *p* < 0.001).

According to the Scheffé test for post-hoc comparisons, there were significant differences between the groups for all items of the questionnaire. Specifically, the results for adolescents with and without LiD differed, indicating that the questionnaire was effective in distinguishing between the two groups.

ANOVA revealed statistically significant differences in the participant groups, responses to the question items, and a significant interaction effect. Significant differences are observed between the subject groups for all items of the questionnaire.

### 3.3. Comparison of Total Scores between Three Selected and All Question Items

The items with the largest score differences between the participant groups across different domains were 3, 12, and 15. Item 3 pertained to speech-in-speech, item 12 to listening effort, and item 15 to psychological effort. Notably, item 15 was excluded from the analysis owing to the significant difference between adolescents without LiD and adults without LiD. Consequently, the analogous psychological item with the largest difference between the adolescents with and without LiD next to item 15, item 14, was selected instead.

The relationships between the scores of each selected item, the aggregate score of the three items, and the total score of all items are depicted in Table 3. Significant correlations were observed between each item and the total set of items for both adolescents with and without LiD; however, the highest correlation was observed with the aggregate score of the three items. This pattern was consistent across both groups.

Figure 1 illustrates a correlation diagram between the aggregate scores of the three items and all items for adolescents with and without LiD. Significant correlations were noted between the total number of items (adolescents without LiD: r = 0.75, *p* < 0.001; adolescents with LiD: r = 0.71, *p* < 0.001). The correlation coefficients for both groups were also significant (r = 0.95, *p* < 0.001). These results indicated that the short questionnaire was effective in screening for LiD.

## 4. Discussion

We examined the utility of the questionnaire for LiD in adolescents both with and without LiD. Subsequently, we compared the results of the questionnaire for LiD between adolescents and adults without LiD (data obtained from a previous study [30]) to examine the utility of the questionnaire in adolescents. The results indicated no significant differences in the scores for each question item between adolescents and adults without LiD [30], except for item 15. This item poses the question, “Do you feel down about asking others to repeat again?” The regret associated with having to request that the speaker repeat statements several times appears to occur regardless of LiD symptoms. As adults tend to be more considerate towards the speaker, the individual score variations for item 15 likely increased. Excluding this item, for junior high school students and older individuals, the response tendencies toward the questionnaire remain consistent. We considered that the listening environment is fundamentally similar for both adolescents and adults and that there is minimal difference in their subjective listening experiences. In line with previous studies [28,30], our results indicate that adolescents can accurately recognize their symptoms and respond to questionnaires effectively.

Recent studies have developed the kid-SSQ [35] and the SSQ-Ch [36] for younger children and adolescents. We employed the SSQ-12, which is typically used for adults, to align with previous studies and investigate differences in response tendencies between adults and adolescents. It is essential to determine whether questionnaires specifically designed for children and adolescents are preferable or if the same questionnaire can be effectively applied to adults. We assessed whether selecting specific items from the questionnaire to screen for LiD symptoms could predict the overall score in adolescents, both with and without LiD. Our findings revealed significant correlations between the short and full questionnaire scores in these groups. Specifically, the total score of the short version correlated more strongly with the full questionnaire than with individual items. This suggests that it is feasible to screen for LiD symptoms using a shortened questionnaire format. The short questionnaire, comprising three items, was simple for respondents to complete and is commonly employed in physical examinations and mental health screenings in educational settings. However, the short questionnaire is intended solely for LiD screening. For a comprehensive assessment, individuals with suspected LiD should complete the full questionnaire, and the results should be analyzed to determine the presence and characteristics of LiD symptoms.

There was continuous distribution of questionnaire scores among adolescents with and without LiD. In developmental disorders, the traits of children with and without these conditions vary widely [37]. Similarly, individuals with LiD may display a range of symptoms. In the present study, adolescents without LiD did not exhibit listening difficulties or low scores. Nevertheless, there could be undetected LiD symptoms within the general population. The broad implementation of straightforward questionnaires could facilitate early identification and intervention for potential LiD symptoms.

Our study identified significant correlations between the short and complete questionnaires designed for adolescents, both with and without LiD. This abbreviated form, comprising three items, could be effectively implemented in physical examinations and mental health assessments within school environments. Early detection and intervention targeting LiD symptoms are recommended. Nonetheless, the study has several limitations, warranting future research. First, further research involving a larger sample size is required to evaluate the practical application of this condensed questionnaire. It is crucial to assess the efficacy of this questionnaire by deploying it across a broad cohort of students during physical examinations or school-based surveys. Second, the appropriate age for questionnaire administration needs further exploration. Specifically, the feasibility of utilizing the questionnaire for elementary school students or the necessity for linguistic adjustments should be evaluated. Third, the correlation between the questionnaire scores and psychological testing outcomes warrants investigation. Understanding the manifestation of actual symptoms in the questionnaire responses, as they relate to psychological test results, is vital. Fourth, the method of data collection also needs to be considered. In this study, we administered the short and complete questionnaires to the same groups of participants. However, the participants will likely answer the questions in a similar way. We should examine comparisons between different groups of participants. Finally, it is important to examine whether similar results can be obtained in other locations and among different ethnic groups. Comparing findings with results from other countries will help to determine the generalizability of the questionnaire.

## 5. Conclusions

We examined the utility of the questionnaire for LiD for adolescents, both with and without LiD. The response patterns of adolescents without LiD to the questionnaire were similar to those of adults with LiD. Furthermore, we investigated whether the choice of questionnaire items used to screen for LiD symptoms in individuals with and without LiD could predict the overall questionnaire score. The results demonstrated that the total score of only three selected questionnaire items was strongly correlated with the overall questionnaire score, facilitating the detection of LiD symptoms. Our study findings suggest that the broad implementation of this simple questionnaire could lead to the early identification of and intervention for potential LiD symptoms.

## Figures and Tables

**Figure 1 children-11-01212-f001:**
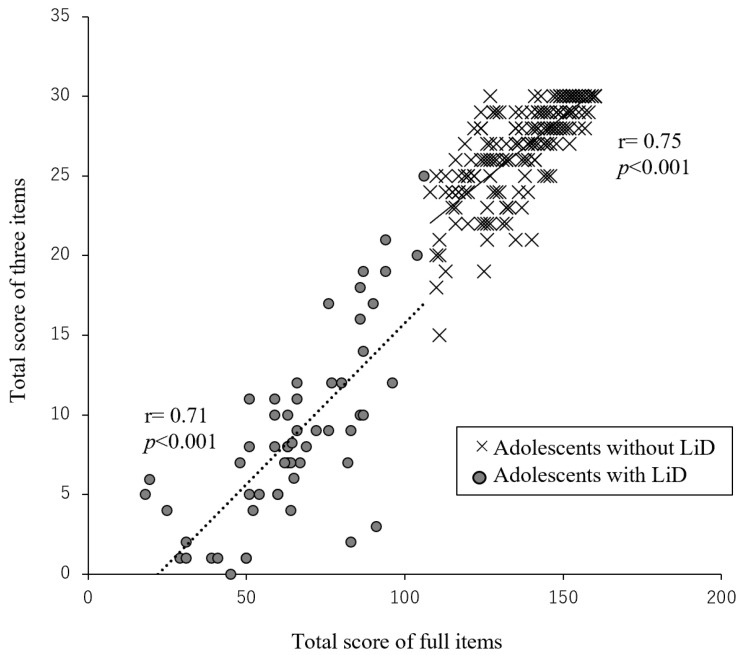
Correlation diagram between total score of three and all items in adolescents with and without LiD. The correlation coefficients for groups with and without LiD are significant (r = 0.95, *p* < 0.001).

**Table 1 children-11-01212-t001:** Mean and standard deviation of each item in the questionnaire for adolescents and adults without LiD.

Question Items	Adolescents without LiD (n = 172)	Adults without LiD (n = 58) *	F-Value	*p*-Value
1	9.18 (1.41)	9.10 (1.24)	0.09	n.s.
2	7.47 (2.46)	7.28 (2.21)	0.56	n.s.
3	8.50 (1.68)	8.35 (1.33)	0.32	n.s.
4	8.64 (1.71)	8.66 (1.53)	0.00	n.s.
5	8.41 (1.76)	8.52 (1.45)	0.17	n.s.
6	8.34 (1.79)	8.38 (1.42)	0.02	n.s.
7	8.30 (1.58)	7.90 (1.61)	2.52	n.s.
8	9.24 (1.04)	9.00 (1.27)	0.77	n.s.
9	8.20 (1.95)	8.02 (2.16)	0.52	n.s.
10	7.46 (2.44)	7.67 (1.93)	0.69	n.s.
11	9.06 (1.30)	9.10 (1.00)	0.00	n.s.
12	8.67 (1.67)	8.43 (1.70)	0.94	n.s.
13	9.64 (0.88)	9.50 (1.13)	0.32	n.s.
14	9.67 (0.85)	9.36 (1.25)	1.44	n.s.
15	8.44 (1.78)	7.64 (1.61)	51.23	*p* < 0.001
16	8.87 (1.78)	8.60 (1.88)	1.12	n.s.

* The results of adults without LiD were extracted from a previous study [23]. There were no significant differences in the scores of each question item between adolescents and adults without LiD [26], except for item 15.

**Table 2 children-11-01212-t002:** Mean and standard deviation of each item in the questionnaire for adolescents with and without LiD.

Question Items	Adolescents without LiD (n = 172)	Adolescents with LiD (n = 54)	F-Value	*p*-Value
1	9.18 (1.41)	5.39 (2.19)	163.71	*p* < 0.001
2	7.47 (2.46)	2.11 (1.72)	322.33	*p* < 0.001
3	8.50 (1.68)	3.00 (2.17)	345.62	*p* < 0.001
4	8.64 (1.71)	3.43 (2.11)	313.69	*p* < 0.001
5	8.41 (1.76)	3.46 (2.09)	276.96	*p* < 0.001
6	8.34 (1.79)	4.96 (2.65)	131.66	*p* < 0.001
7	8.30 (1.58)	5.37 (2.69)	98.99	*p* < 0.001
8	9.24 (1.04)	7.28 (2.30)	49.87	*p* < 0.001
9	8.20 (1.95)	3.81 (2.51)	221.33	*p* < 0.001
10	7.46 (2.44)	5.91 (2.78)	29.98	*p* < 0.001
11	9.06 (1.30)	6.24 (2.65)	92.63	*p* < 0.001
12	8.67 (1.67)	2.48 (2.26)	453.01	*p* < 0.001
13	9.64 (0.88)	4.24 (3.25)	357.94	*p* < 0.001
14	9.67 (0.85)	3.37 (3.39)	478.44	*p* < 0.001
15	8.44 (1.78)	2.15 (3.27)	478.25	*p* < 0.001
16	8.87 (1.78)	3.63 (3.31)	332.52	*p* < 0.001

**Table 3 children-11-01212-t003:** Correlation coefficient between each item or a total of three items and all items in children with and without LiD.

Items	Adolescents without LiD	Adolescents with LiD
No. 3 You are in conversation with one person in a room where there are many other people talking. Can you follow what the person you are talking to is saying?	0.67 **	0.58 **
No. 12 Do you have to concentrate very much when listening to someone or something?	0.51 **	0.51 **
No. 14 Do you think that you want to be alone because of your listening difficulties?	0.32 **	0.57 **
Total three items	0.75 **	0.71 **

** *p* < 0.001 Significant correlations of the total scores between the selected three items and all items are observed.

## Data Availability

The data that support the findings of this study are available from the corresponding author upon reasonable request.

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
