# Peer review of "Utility of a Questionnaire Short Form for Adolescents with Listening Difficulties in Japan"

_children, 2024, doi:10.3390/children11101212_

Round 1
Reviewer 1 Report
Comments and Suggestions for Authors
11. It is unclear how the authors use the term “Adaptation” normally in the context of questionnaires, it refers to changes or modifications of items or an entire questionnaire to create a version which suits a cultural or specific context. Was the questionnaire used the same as the previous study, or was it modified? If it was, the authors should describe the modifications. Still, if the questionnaire were not modified for adolescents, I would not use that term as it can generate confusion.
Could you explain the short form of the questionnaire? To my understanding, it is SSQ12+4 items. Do the 16 items refer to a short form? Which was the “original form”, the one of ref 25?
2. Could you provide more information about the questionnaire, such as the maximum points (160 points?), so the readers can understand it?, Which was the cut-off criterion?
3. Could you explain how the “listening difficulty” was defined (criterion) regarding participant recruitment?
4. The questionnaire was self-administered? Or as an interview?
5. The methodology description should be improved, mainly the questionnaire descriptions
6. Should we assume that there is no significant age difference between the LD and non-LD groups (adolescents)?
7. Pag 4; Results: Could you explain what was the interaction effect found?
8. The correlation of the 3 items with the total score would be enough to predict results?. A strong correlation doesn’t mean causation, only associations. What about content validity or discriminative validity evaluation? Faced validity and reliability were formally assessed?
9. Page 7: Revise the correct use of “adaptation” in the manuscript. It is unclear if the original questionnaire was modified to be used in the group.
10. Page 8: discussion: the word “lowly” is appropriate?
11. The title stated, “validation of….” Would the comparisons between groups and the questions' correlations be enough to validate the questionnaire?
12. The discussion lacks some explanations of the results, more so than the explanation for item 15. Providing some brief insights into such differences would be valuable if comparisons are conducted. Is it possible that listening environment differences between adults and adolescents could account for that difference?
13. Could you clarify this sentence from the conclusion “ The tendency of adolescents to respond to the questionnaire mirrored that of adults”. Which adolescents? Which group?
Comments on the Quality of English Language
minor revisons
Author Response
Point-by-point responses to the Reviewers’ comments
We thank the reviewer for the time and effort dedicated to reviewing our manuscript. We sincerely appreciate the thoughtful suggestions and comments, which have greatly helped us to improve the manuscript. We have carefully considered these comments, responded to them below in a point-by-point manner, and revised the manuscript accordingly. All revisions are highlighted in yellow in the main manuscript. We trust that your comments have been addressed.
1) It is unclear how the authors use the term “Adaptation” normally in the context of questionnaires, it refers to changes or modifications of items or an entire questionnaire to create a version which suits a cultural or specific context. Was the questionnaire used the same as the previous study, or was it modified? If it was, the authors should describe the modifications. Still, if the questionnaire were not modified for adolescents, I would not use that term as it can generate confusion.”
Response: To avoid confusion, we have accordingly deleted the word “adaptation” and replaced it with the word “utility” throughout the manuscript.

2) Could you explain the short form of the questionnaire? To my understanding, it is SSQ12+4 items. Do the 16 items refer to a short form? Which was the “original form”, the one of ref 25?”
Response: We apologize for the confusion and oversight. To avoid any confusion, we now refer to the three-item questionnaire as the “short form” and have not applied the same term to other versions. The SSQ-12 is a selected 12-item questionnaire of the SSQ. In addition to these 12 items, we incorporated 4 from the 2002 Questionnaire on Hearing. We then screened items that were significantly different between adolescents with and without LiD, resulting in the three-item questionnaire, referred to as “short form.”
3) Could you provide more information about the questionnaire, such as the maximum points (160 points?), so the readers can understand it?, Which was the cut-off criterion?
Response: We have accordingly added the maximum point and cut-off criterion to the text (page 4, line 150-151, 154). The 16 items are scored on a scale ranging from 0 to 10, with responses varying from "not at all" to "perfect." The total scores ranged from 0 to 160. The cutoff value for LiD in the questionnaire was ≤109.
4) Could you explain how the “listening difficulty” was defined (criterion) regarding participant recruitment?
Response: Following your comment, we have added the inclusion criteria for the participants in the text (page 3, line 108-115). Briefly, all participants met the diagnostic criteria for LiD in Japan and were either referred to or joined the peer support organization on their own. Adolescents without LiDs were selected from cooperating schools. Based on prior research, participants were limited to junior and senior high school students who reported no hearing issues during their annual physical exams and were unaware of any LiD. After obtaining informed consent, they completed the questionnaire. We included responses from students who had answered all questions without missing any.
5) The questionnaire was self-administered? Or as an interview?
Response: We apologize for the lack of information. The questionnaire was self-administered and required individual responses. The participants were asked to answer each item according to their listening status. We have now clarified this in the text. (page 4, line 158-160)
6) The methodology description should be improved, mainly the questionnaire descriptions
Response: We apologize for the lack of description of the questionnaires. As addressed in the comments above, we have now described the inclusion criteria of the participants, how the 3-item questionnaire was developed, and that the questionnaire was self-administered. Please refer to the revised Methods section.
7) Should we assume that there is no significant age difference between the LD and non-LD groups (adolescents)?
Response: Indeed, there was no significant age difference between the groups. We have now also reported the lack of significant differences in the responses between grades in the results section (page 4, line 186-187).
8) Pag 4; Results: Could you explain what was the interaction effect found?
Response: We have accordingly added an explanation of the interaction effect (page 4, line 183-185).
9) The correlation of the 3 items with the total score would be enough to predict results?. A strong correlation doesn’t mean causation, only associations. What about content validity or discriminative validity evaluation? Faced validity and reliability were formally assessed?
Response: Thank you for your suggestion. We selected three question items for screening LiD. However, if there are indications of suspected LiD, we plan to administer the full version of the questionnaire. We added this clarification to the main text (page 8, line 271-274).
10) Page 7: Revise the correct use of “adaptation” in the manuscript. It is unclear if the original questionnaire was modified to be used in the group.
Response: As mentioned in our response to your first comment, we replaced the word “adaptation” with “utility” throughout the text.
11) Page 8: discussion: the word “lowly” is appropriate?
Response: We apologize for this oversight and have replaced the word “lowly” with “low scores”.
12) The title stated, “validation of….” Would the comparisons between groups and the questions' correlations be enough to validate the questionnaire?
Response: We replaced the word “validity” with “utility” in the title to address your comment.
13) The discussion lacks some explanations of the results, more so than the explanation for item 15. Providing some brief insights into such differences would be valuable if comparisons are conducted. Is it possible that listening environment differences between adults and adolescents could account for that difference?
Response: In response to your comment, we explained that no major differences in the listening environment and participant feelings were noted between the groups, except for item 15.
14) Could you clarify this sentence from the conclusion “ The tendency of adolescents to respond to the questionnaire mirrored that of adults”. Which adolescents? Which group?
Response: We apologize for this unclear sentence: “The response patterns of adolescents without LiD to the questionnaire were similar to those of adults with LiD.”
Reviewer 2 Report
Comments and Suggestions for Authors
Abstract:
- An excessive amount of detail and length are present in the abstract. It ought to be condensed into one or two paragraphs that emphasize the main ideas.
- Eliminate particulars such as the quantity of subjects and the items on the questionnaire. Just provide a high-level summary of the main findings.
- Provide a more succinct explanation of the study's primary goal and conclusions.
Introduction:
- Needs to give additional background information on listening impairments and current surveys. Why is a shorter survey required? What are the present tools' limitations?
- genetic auditory syndromes present diffent phenotypes and should be influence outcomes. discuss and cite doi:10.3390/biomedicines11061616
- It is possible to clarify the study's purpose and justification more precisely.
Methods:
- Explain how a questionnaire is developed. How were particular items chosen and put together? Give further specifics.
-The criteria and sampling strategy for teenage groups has to be clarified.
The statistical analysis section needs to provide more information about the particular tests that were employed and their justifications.
Results:
- Use headings to present the main conclusions in a more logical order.
- Minimize superfluous information and tables. Describe the main findings in two to three tables or figures.
- Don't merely provide the numbers; instead, interpret and explain the outcomes.
Discussion:
- Should center on analyzing the importance and ramifications of the study's findings. Extend this section.
- It is important to talk about the study's limitations and the need for more research.
Clarity needs to be increased in the writing style; too many details make the text hard to read.
only few grammatical errors were found in the article.
Author Response
Point-by-point responses to the Reviewers’ comments
We thank the reviewer for the time and effort dedicated to reviewing our manuscript. We sincerely appreciate the thoughtful suggestions and comments, which have greatly helped us to improve the manuscript. We have carefully considered these comments, responded to them below in a point-by-point manner, and revised the manuscript accordingly. All revisions are highlighted in yellow in the main manuscript. We trust that your comments have been addressed.
1) Abstract:
- An excessive amount of detail and length are present in the abstract. It ought to be condensed into one or two paragraphs that emphasize the main ideas.
- Eliminate particulars such as the quantity of subjects and the items on the questionnaire. Just provide a high-level summary of the main findings.
- Provide a more succinct explanation of the study's primary goal and conclusions.
Response: We have accordingly revised the entire abstract, ensuring that it is concise. Kindly refer to the revised abstract in the main text.
2) Introduction:
- Needs to give additional background information on listening impairments and current surveys. Why is a shorter survey required? What are the present tools' limitations?
- genetic auditory syndromes present diffent phenotypes and should be influence outcomes. discuss and cite doi:10.3390/biomedicines11061616
- It is possible to clarify the study's purpose and justification more precisely.
Response: We have accordingly added the rationale of the study, highlighting the limitations of the currently available tools, and the aims of the study.
We also mentioned genetic auditory syndromes, citing a new study in the introduction section. However, as our study focuses on screening for listening difficulties, we believe that genetic auditory syndromes do not significantly influence the results.
3) Methods:
- Explain how a questionnaire is developed. How were particular items chosen and put together? Give further specifics.
-The criteria and sampling strategy for teenage groups has to be clarified.
The statistical analysis section needs to provide more information about the particular tests that were employed and their justifications.
Response: In accordance with your comment, we added the detailed information of the questionnaire, the sampling strategy, and statistical justifications to the text.
4) Results:
- Use headings to present the main conclusions in a more logical order.
- Minimize superfluous information and tables. Describe the main findings in two to three tables or figures.
- Don't merely provide the numbers; instead, interpret and explain the outcomes.
Response: We have accordingly divided the results section into headings. We also revised the text and explained the results. Kindly refer to the revised results section.
5) Discussion:
- Should center on analyzing the importance and ramifications of the study's findings. Extend this section.
- It is important to talk about the study's limitations and the need for more research.
Clarity needs to be increased in the writing style; too many details make the text hard to read.
Response: We have carefully revised the discussion, ensuring that it is clear. We have also mentioned the limitations of our study and the scope for future research.
Reviewer 3 Report
Comments and Suggestions for Authors
This paper presents useful evidence on an important topic. There are some suggestions for improvement:
1) The Introduction should describe more research on specifically LiD in adolescents, as that is the focus of the study.
2) Example items should be given for each subscale of the SSQ-12 and the 2002 Questionnaire on hearing. The scaling of the SSQ-12 should be noted. A coefficient alpha should be given for the SSQ-12 for each subscale.
3) There should be demographic information given on the genders of the participants and what percentages were in each grade.
4) In the first paragraph of the Results, what was the correlation between age and the total questionnaire for adults?
5) T-tests comparing both adolescent groups on the total score or subscale scores for the hearing measure should have been conducted, not solely item analyses, to see if there were overall differences in responding. The two-group item comparisons could have been done with paired t-tests and not F tests, as they were two-group comparisons.
6) In the second-to-last paragraph of the Discussion, it states that there was "consistency" in questionnaire scores between adolescent groups. This is unclear, as there were significant differences between the two groups on multiple items.
7) The limitations of generalizability to other geographic locations and ethnicities as well as shared method variance, as both questionnaires were filled out by the same participants, should be mentioned.
Author Response
Point-by-point responses to the Reviewers’ comments
We thank the reviewer for the time and effort dedicated to reviewing our manuscript. We sincerely appreciate the thoughtful suggestions and comments, which have greatly helped us to improve the manuscript. We have carefully considered these comments, responded to them below in a point-by-point manner, and revised the manuscript accordingly. All revisions are highlighted in yellow in the main manuscript. We trust that your comments have been addressed.
1) The Introduction should describe more research on specifically LiD in adolescents, as that is the focus of the study.
Response: In accordance with your comment, we added the detailed information of the questionnaire, the sampling strategy, and statistical justifications to the text.
2) Example items should be given for each subscale of the SSQ-12 and the 2002 Questionnaire on hearing. The scaling of the SSQ-12 should be noted. A coefficient alpha should be given for the SSQ-12 for each subscale.
Response: We have accordingly included example items for each subscale of the SSQ-12 and the 2002 Questionnaire on Hearing in the Methods section. Moreover, we have noted the scaling used for the SSQ-12. The validity of the SSQ-12 was examined in a previous study [30], and that of the LiD questionnaire was assessed in another study [26].
3)There should be demographic information given on the genders of the participants and what percentages were in each grade.
Response: While we collected age data for the participants, we did not collect information on their grades. Therefore, we provided the mean and standard deviation for age. Furthermore, we were unable to accurately report the male-to-female ratio because some participants did not specify their gender. However, as the participants without LiD enrolled were from a coeducational school, we believe that gender did not have a significant impact on our findings.
4) In the first paragraph of the Results, what was the correlation between age and the total questionnaire for adults?
Response: Thank you for your suggestion. In this study, we focused on comparing adolescents with and without LiD. The data for adults without LiD were obtained from previous research [26] and were used only for comparison with adolescents. Therefore, we did not conduct a detailed analysis of adult data, including the correlation between age and the total questionnaire score for adults.
5) T-tests comparing both adolescent groups on the total score or subscale scores for the hearing measure should have been conducted, not solely item analyses, to see if there were overall differences in responding. The two-group item comparisons could have been done with paired t-tests and not F tests, as they were two-group comparisons.
Response: In this study, we conducted ANOVA to examine overall trends, items, and interactions. We also conducted a t-test for comparison between items; however, as the results were similar, we focused on the post-hoc comparisons for more detailed analysis.
6) In the second-to-last paragraph of the Discussion, it states that there was "consistency" in questionnaire scores between adolescent groups. This is unclear, as there were significant differences between the two groups on multiple items.
Response: We apologize for using this word and have revised it to “continuous distribution of questionnaire scores.”
7) The limitations of generalizability to other geographic locations and ethnicities as well as shared method variance, as both questionnaires were filled out by the same participants, should be mentioned.
Response: We have accordingly revised the limitations of our study in the discussion section to include the points that you have raised.
Round 2
Reviewer 1 Report
Comments and Suggestions for Authors
Not applicable
Author Response
We thank the reviewer for the time and effort dedicated to reviewing our manuscript.
Reviewer 3 Report
Comments and Suggestions for Authors
Several reviewer comments were not addressed:
1) No literature on LiD in adolescents was added to the Introduction.
2) Coefficient alphas were not given for the SSQ-12 subscales from the study. Previous information is not sufficient, as data needs to be reported on reliability for the current sample.
3) T-tests were not conducted comparing both groups of adolescents on total or subscale scores. The paper remained focused on item comparisons.
4) The shared method variance limitation was not added to the text.
Author Response
Point-by-point responses to the Reviewers’ comments
We thank the reviewer for the time and effort dedicated to reviewing our manuscript. We sincerely appreciate the thoughtful suggestions and comments, which have greatly helped us to improve the manuscript. We have carefully considered these comments, responded to them below in a point-by-point manner, and revised the manuscript accordingly. All revisions are highlighted in yellow in the main manuscript. We trust that your comments have been addressed.
- No literature on LiD in adolescents was added to the Introduction.
We apologize for the inadequate explanation. Some reference cited in the introduction examines for adolescents with and without LiD (APD). I have added this to make it clear. However, most of the research on LiD has been conducted in school-aged children. We also noted that there are few studies on adolescents.
- Coefficient alphas were not given for the SSQ-12 subscales from the study. Previous information is not sufficient, as data needs to be reported on reliability for the current sample.
We have accordingly added the results of Cronbach’s alpha coefficient of questionnaire (total items, SSQ-12, and psychological domain) in this study. (In recent papers, the SSQ-12 is not analyzed by subscale, so in this study, it is also presented as a total score) And the results of Cronbach’s alpha coefficient of SSQ-49 in the previous study have also added in the introduction.
- T-tests were not conducted comparing both groups of adolescents on total or subscale scores. The paper remained focused on item comparisons.
We have accordingly added the results of t-test for comparing the participant groups.
4) The shared method variance limitation was not added to the text.
We apologize for our insufficient description. Following your comment, we have added the methodological problem we administered the compared questionnaires to the same participant groups.